# Pharmacodynamic Effects of Standard versus High Caffeine Doses in the Developing Brain of Neonatal Rats Exposed to Intermittent Hypoxia

**DOI:** 10.3390/ijms22073473

**Published:** 2021-03-27

**Authors:** Kutilda Soontarapornchai, Charles L. Cai, Taimur Ahmad, Jacob V. Aranda, Ivan Hand, Kay D. Beharry

**Affiliations:** 1Department of Pediatrics, Division of Neonatal-Perinatal Medicine, State University of New York, Downstate Medical Center, Brooklyn, NY 11203, USA; drkutilda@gmail.com (K.S.); Charles.cai@downstate.edu (C.L.C.); taahmad@alumni.emory.edu (T.A.); Jacob.aranda@downstate.edu (J.V.A.); Ivan.Hand@nychhc.org (I.H.); 2Department of Ophthalmology, State University of New York, Downstate Medical Center, Brooklyn, NY 11203, USA; 3State University of New York Eye Institute, Brooklyn, NY 11203, USA; 4Department of Pediatrics, Division of Neonatal-Perinatal Medicine, Kings County Hospital Center, Brooklyn, NY 11203, USA

**Keywords:** apoptosis, caffeine, cerebral cortex, hyperoxia, mylenation, neonatal intermittent hypoxia, oxidative stress

## Abstract

(1) Background: Caffeine citrate, at standard doses, is effective for reducing the incidence of apnea of prematurity (AOP) and may confer neuroprotection and decrease neonatal morbidities in extremely low gestational age neonates (ELGANs) requiring oxygen therapy. We tested the hypothesis that high-dose caffeine (HiC) has no adverse effects on the neonatal brain. (2) Methods: Newborn rat pups were randomized to room air (RA), hyperoxia (Hx) or neonatal intermittent hypoxia (IH), from birth (P0) to P14 during which they received intraperitoneal injections of LoC (20 mg/kg on P0; 5 mg/kg/day on P1-P14), HiC (80 mg/kg; 20 mg/kg), or equivalent volume saline. Blood gases, histopathology, myelin and neuronal integrity, and adenosine receptor reactivity were assessed. (3) Results: Caffeine treatment in Hx influenced blood gases more than treatment in neonatal IH. Exposure to neonatal IH resulted in hemorrhage and higher brain width, particularly in layer 2 of the cerebral cortex. Both caffeine doses increased brain width in RA, but layer 2 was increased only with HiC. HiC decreased oxidative stress more effectively than LoC, and both doses reduced apoptosis biomarkers. In RA, both caffeine doses improved myelination, but the effect was abolished in Hx and neonatal IH. Similarly, both doses inhibited adenosine 1A receptor in all oxygen environments, but adenosine 2A receptor was inhibited only in RA and Hx. (4) Conclusions: Caffeine, even at high doses, when administered in normoxia, can confer neuroprotection, evidenced by reductions in oxidative stress, hypermyelination, and increased Golgi bodies. However, varying oxygen environments, such as Hx or neonatal IH, may alter and modify pharmacodynamic actions of caffeine and may even override the benefits caffeine.

## 1. Introduction

Apnea of prematurity (AOP) commonly occurs in extremely low gestational age neonates (ELGANs) with gestational ages of <28 weeks. AOP is related to instability of respiratory control and results in intermittent hypoxia (IH). Neonatal IH is defined as brief, repetitive cycles of arterial oxygen desaturations followed by re-oxygenation [1,2,3,4,5]. An IH event is usually defined as a decline in SaO_2_ by 5% lasting <3 min in duration [2,3,4,5]. Re-oxygenation following an IH event can occur in normoxia or hyperoxia (Hx). Prolonged apnea lasting more than 20 s may have detrimental effects on cerebral hemodynamics, and is associated with adverse neurological and cognitive outcomes [6,7]. Animal models of neonatal IH also show long-term cognitive deficits, white matter injury and hypomyelination [8,9,10]. ELGANs are particularly vulnerable to oxidative stress due to immature antioxidant systems and exposure to supplemental oxygen [2,11,12], resulting in adverse neurological outcomes [13,14].

Energy demands in the brain are high, requiring a continuous supply of O_2_. Chronic IH and oxygen deprivation in the brain leads to reduction in energy in the form of adenosine triphosphate (ATP) and brain ischemia [15], followed by membrane depolarization and an uncontrolled influx of intracellular Ca2^+^ and a glutamate surge [16,17]. In hypoxia, adenosine increases up to 100-fold from extracellular ATP breakdown and adenosine extrusion from ischemic cells [18,19,20]. Adenosine, acting via its receptors, A1R, A2AR, A2BR and A3R, mediate opposing inhibitory and excitatory effects. In particular, A2AR is induced, in part, via hypoxia inducible factor (HIF) action [21] in conditions of hypoxia and neuroinflammation [22,23]. Indeed, our laboratory showed that the adenosine receptor antagonist 8-phenyltheophylline substantially reduced hypoxia-induced cerebral vasodilatation in newborn piglets [24]. Other studies show that HIF_1α_ itself can induce microglia cell death [25].

Caffeine, a methylxanthine universally used for prevention or treatment of AOP [26,27,28,29,30] exerts its pharmacologic actions by non-selectively inhibiting adenosine A1 and A2 receptors [31], which increases neuronal firing, releases brain norepinephrine, dopamine, and serotonin [32], and has been shown to be neuroprotective [33]. The randomized placebo-controlled mega trial on Caffeine for Apnea of Prematurity (CAP) trial demonstrated beneficial effects on major neonatal morbidities and significant improvements in long term outcomes [28]. In that study, 2006 ELGANs (birthweight 500–1250 g) were randomly assigned to standard caffeine doses consisting of 20 mg/kg loading/5 mg/kg/day maintenance) or placebo for two weeks. Caffeine reduced the incidence of cerebral palsy and cognitive impairment and significantly improved gross motor function [29,30]. Caffeine has been shown to suppress HIF_1α_ [34,35], which may contribute to its neuroprotection in oxygen deprivation. The effects of caffeine are dose-dependent and at low doses, caffeine antagonizes the action of adenosine A1R, A2AR, and A2BR [36]. While high caffeine doses resulting in plasma levels of 192 or 567 mg/L can produce severe adverse effects and death in adults [37], accidental caffeine overdose in premature infants resulting in plasma levels as high as 217.5 mg/L [38] and cerebrospinal fluid caffeine levels of 115 mg/L [39] were tolerated with transient side effects. Animal studies and neonatal clinical trials on high caffeine doses have reported adverse molecular and cellular effects on the developing brain, increased cerebral injury, and abnormal motor performance [40,41]. In contrast, studies report that high doses administered to preterm infants decrease failure to extubate and duration of mechanical ventilation [42]. At high doses caffeine may interact with many molecular targets producing undesired pharmacologic effects, particularly in ELGANs who are sensitive to supra-pharmacologic doses of caffeine [43]. Thus, the effects of high caffeine doses on the neonatal brain when administered in neonatal IH remains to be determined.

The preterm white matter is highly vulnerable to oxidative damage and, especially, lipid peroxidation-mediated injury [44]. F-isoprostanes, particularly 8-isoPGF_2α_, are the most reliable biomarkers for determination of oxidative stress and lipid peroxidation in human cerebral tissue [45,46,47]. However, 8-iso-PGF_2α_ is a prostaglandin-like product formed from oxidation of arachidonic acid and phospholipids. Generally, they are produced chemically, a process that is independent of the cyclooxygenase enzyme. However, it has been recently shown that 8-iso-PGF_2α_ is also produced via enzymatic lipid peroxidation [48]. Therefore, distinguishing the contribution of each pathway is crucial for correct interpretation of the biomarker. The use of the ratio of 8-iso-PGF_2α_ to PGF_2α_ differentiates between enzymatic and chemical lipid peroxidation. Data from our laboratory as well as others, show that neonatal IH activates reactive oxygen species (ROS) formation [49,50,51], which in excess results in oxidative stress, DNA-fragmentation, and cell death via activated caspase cascade [52,53]. We also showed that caffeine treatment alone, and in combination with non-steroidal anti-inflammatory drugs (NSAIDs), preserved astrocyte integrity [54], and promoted retinal neural development [55], in neonatal IH. Therefore, based on our previous findings and the known anti-inflammatory, anti-apototic, antioxidant, and anti-proliferative properties of caffeine in hypoxia-induced neonatal brain injury [56,57], we tested the hypothesis that high caffeine doses confer no significant adverse effects in the brain of neonatal rats exposed to IH. The overarching objective was to determine whether neonatal IH alters the neuroprotective actions of caffeine. The specific objectives were: (1) to compare the effects of standard versus high caffeine doses on IH-induced brain growth and histopathology; (2) to determine whether high doses of caffeine influences IH-induced biomarkers of oxidative stress, DNA damage, and apoptosis in the neonatal brain; and (3) to examine the inhibitory effects of high caffeine doses on brain adenosine receptors in neonatal IH. The primary outcomes were histopathology and myelination, and the secondary outcomes were morphometry, oxidative stress and apoptosis biomarkers.

## 2. Results

### 2.1. Blood Gases

Mixed arterial-venous blood gases on the day of sacrifice are presented in Table 1. In RA, both caffeine doses resulted in blood gases that were comparable with placebo saline. Exposure to Hx produced the most changes in all parameters and compared to LoC. HiC resulted in lower pH, higher PCO_2_ and lower PO_2_ values. These effects were reversed in neonatal IH.

### 2.2. Growth

Growth parameters are presented in Figure 1. Panels A and B shows the percentage changes in body weight and body length, respectively. Mean body weights at birth were 7.0 ± 0.16 (placebo saline); 6.9 ± 0.14 (LoC); and 6.9 ± 0.11 (HiC). Body weight accretion was lower in Hx and neonatal IH and was increased and mproved with LoC, but decreased with HiC. Similarly, body length accretion was significantly decreased in response to Hx and neonatal IH, and was slightly corrected with HiC treatment in Hx. Organ weight can be one of the most sensitive indicators of effects of an experimental intervention and brain to body weight ratios are more important indicators of brain growth than actual brain weights [58,59]. Panels C and D shows the total brain weight and brain to body weight ratios, respectively. In Hx, brain weight was lower with LoC treatment in Hx and neonatal IH compared to RA, while HiC caused a significant substantial reduction in brain weight when administered in neonatal IH (Figure 1, panel C). Panel D shows brain to body weight ratios wherein LoC treatment in Hx and neonatal IH caused reductions compared to placebo saline in RA. In contrast, a dose-dependent and opposing effect was noted for LoC which increased brain/body weight ratios whereas HiC significantly decreased it. 

### 2.3. Histopathology

Figure 2 shows representative images of the H&E stained cerebral cortex. RA groups are presented on the top panels, Hx groups are presented in the middle panels, and neonatal IH groups on the bottom panels. Placebo saline groups are positioned on the left, LoC in the middle and HiC on the right. Groups exposed to placebo saline in RA showed no abnormalities and layers were clearly defined. Hx resulted in loss of cells in the external granular layer (layer II) with reduced distinction of the layers. Neonatal IH resulted in severe hemorrhage (arrow), but with less reduction in layer II stellate cells. LoC treatment preserved cell number and layering, but treatment in Hx resulted in widening of the molecular layer (layer I) with reduced horizontal cells, and reduction in pyramidal cells in the external pyramidal layer (layer III). Treatment with LoC in neonatal IH reduced layer I width, but the effect of reduced pyramidal cells in layer III persisted. There was also mild evidence of apoptosis and hemorrhage. In RA, there was evidence of apoptosis and hemorrhage (arrows, although no significant effects on cell number or layer distinction was observed. Apoptosis and hemorrhage persisted and worsened with treatment in Hx and neonatal IH (arrows). Table 2 presents morphometric analyses of the cerebral cortex. Total brain width increased with both doses in RA and with LoC in Hx. Due to reductions in ability to distinguish the layers, only layers I and II were measured. In RA, layer II increased 2-fold with HiC. Under conditions of Hx, LoC, but not HiC increased brain weights and layers I and II. The effects of neonatal IH or caffeine predominated in layer II with significant increases compared to placebo saline in RA.

Other histopathological findings (Appendix A) include damage in the dentate gyrus (DG), CA-1 and CA-3, manifested as perivascular and perineuronal vacuolations (a bubble-liked cytoplasm with dark and dense nucleus) and apoptotic bodies (slit-like membrane-surrounded cytoplasm), characteristics of neurodegeneration. CA-3 also showed multiple pyknotic neurons (shrunken and angular neurons with condense nucleus), which is another feature of neuronal death. The inter-neuronal alignments in all three subregions were loose and distorted. In the LoC group exposed to RA, the hippocampal subregions appeared normal and similar to controls. LoC treatment in Hx reduced pyknosis and vacuolated neurons observed in Sal-treated groups exposed to Hx. LoC treatment in IH showed no neuronal death or degeneration in the hippocampus. HiC treatment in RA induced hippocampal neuronal necrosis, degeneration, and disorganization, similar to Sal treatment in Hx. This damage occurred mainly in the dentate gyrus and CA-3 regions. Similar findings were noted with HiC treatment in Hx. However, HiC treatment in IH resulted in minimal changes and no appearances of neurodegeneration (Appendix A). Hemorrhage was also noted in the choroid plexus particularly in the groups exposed to neonatal IH (Appendix A).

### 2.4. Oxidative Stress and Apoptosis

Figure 3 shows oxidative stress and apoptosis biomarkers. Oxidative stress in the cerebral cortex homogenates is evidenced by 8-isoPGF_2α_ (panel A) and oxidative DNA damage is evidenced by 8-OHdG (panel B). Exposure to Hx increased 8-isoPGF_2α_ levels in the placebo saline and LoC groups. HiC was more effective for than LoC in suppressing oxidative stress. 8-OHdG increased in the placebo saline group exposed to Hx and neonatal IH, and in response to both caffeine doses, although the levels were higher in Hx and neonatal IH than RA. Caspase-3 and caspase-9 levels in the brain homogenates are presented in panel C and D, respectively. While caspase-3 was reduced in Hx and neonatal IH in all treated groups, caspase-9 was elevated in all groups exposed to neonatal IH. 8-isoPGF_2α_ can be produced enzymatically via cyclooxygenase or chemically. To determine its source, levels of PGF_2α_ and the ratio of 8-isoPGF_2α_/PGF_2α_ was determined and presented in Figure 4. Panel A shows that brain PGF_2α_ levels were equipotently decreased with both doses of caffeine. The ratios of 8-isoPGF_2α_/PGF_2α_ were higher with LoC in Hx than IH (panel B).

### 2.5. Immunoreactivity

Representative caspase-3, caspase-9, A_1_R and A_2A_R immunoreactivity are presented in Figure 5, Figure 6, Figure 7 and Figure 8. Caspase-3 immunoreactivity is presented in Figure 5 (red). Nuclei are stained with DAPI (blue). Caspase-3 decreased in both Hx and IH environments. Treatment with placebo saline resulted in minimal caspase-3 immunoreactivity. In contrast, treatment with LoC and HiC resulted in more robust caspase-3 staining in all oxygen environments.

Caspase-9 immunoreactivity was higher in the placebo saline group exposed to Hx and neonatal IH, but was lower with LoC in Hx. The highest reactivity was seen in the HiC group exposed to neonatal IH (Figure 6). 

A_1_R immunoreactivity is presented in Figure 7. A_1_R was generally expressed around the nucleus of the neurons in layer II, as well as in Layer I of the cerebral cortex in the placebo controls. Both Hx and IH resulted in higher A_1_R expression. Treatment with LoC in RA and IH decreased A_1_R expression, while treatment in Hx resulted in a mild increase. Similarly, treatment with HiC in RA and Hx decreased A_1_R expression predominantly in layer I of the cerebral cortex, however, expression robustly increased in the deeper layers in the RA group. Treatment in neonatal IH also resulted in elevated A_1_R. 

Immunoreactivity of A_2A_R in the cerebral cortex is presented in Figure 8. A_2A_R was highly expressed in the placebo controls exposed to neonatal IH. Treatment with LoC decreased A_2A_R expression in all oxygen environments, as did treatment with HiC in RA and Hx. Treatment with HiC in neonatal IH was elevated particularly in the hemorrhagic areas. 

Figure 9 shows the quantitative analysis of stain intensities for caspase-3 (panel A), caspase-9 (panel B), A_1_R (panel C), and A_2A_R (panel D) receptors. Data are mean ± SEM (*n* = 12 measurements/group). Caspase-increased in the Hx and neonatal IH groups treated with placebo saline, with the highest elevation noted with neonatal IH exposure. Both groups of caffeine significantly decreased caspase-3 in all environments, however, HiC was less effective in neonatal IH. Similarly, both caffeine doses suppressed caspase-9 in RA and Hx, but the effect was abolished in neonatal IH. This was most likely due to lower levels in the placebo saline group. A similar response was seen with the adenosine receptors. 

### 2.6. Myelination 

Representative myelin and Golgi stains are presented in Figure 10 and Figure 11 respectively. Treatment with LoC and HiC showed hypermyelination in RA compared to placebo saline. Conversely, all groups showed hypomyelination in response to Hx and neonatal IH. However, of those groups, the ones treated with HiC tended to have better myelination than placebo saline and LoC. This was further demonstrated in Figure 12 which shows the myelin stain quantification. 

Representative Golgi stains are presented in Figure 11 (images are 10× magnification, scale bar = 100 µm). Image analysis to determine the mean number of Golgi stained neurons was conducted at 40× magnification (20 measurements/group). In the placebo saline group, exposure to neonatal IH resulted in decreased numbers of Golgi bodies (17.7 ± 0.36, *p* < 0.01) compared to RA (27.2 ± 0.29) and Hx (22.0 ± 0.43). LoC treatment in RA increased the numbers, but this effect was dramatically reduced with treatment in Hx (2.5 ± 0.3, *p* < 0.01) and neonatal IH (8.9 ± 0.26, *p* < 0.01). Treatment with HiC in RA also increased Golgi stained neurons compared to placebo saline, as did treatment in neonatal IH (24.4 ± 0.23). Although the numbers declined with treatment in Hx (15.2 ± 0.34, *p* < 0.01), they were higher than LoC treatment in Hx. 

Table 3 shows the morphologic parameters of the golgi bodies. Both caffeine doses in RA increased the total neurons and number of dendrites per neuron. This effect persisted in Hx and IH. Dendritic length was decreased with HiC in RA, but not in Hx or IH. 

## 3. Discussion

The present study is the first to examine and compare the pharmacodynamic effects of standard and high doses of caffeine in the neonatal brain during hyperoxia and intermittent hypoxia. We used high doses that were four times higher than standard of care. These high doses are currently being used in ELGANs for AOP and for reducing extubation failure, respectively. We utilized a reliable and reproducible IH paradigm that simulates frequent arterial oxygen desaturations experienced by ELGANs. This model, which targets those preterm infants who experience frequent IH with resolution in hyperoxia between episodes, has repeatedly been shown to produce oxidative stress in neonatal rats [60,61]. The major findings are: (1) treatment with LoC increases brain growth which is nullified by HiC. In Hx and neonatal IH, HiC causes reductions in weight accretion and brain/body weights in neonatal IH. Whether this resolves post treatment with catchup growth remains to be determined. Failure to catch-up may lead to adverse neurological outcomes. It appears that there may be a critical dose of caffeine where caffeine protection is outweighed by increased metabolic rate and cell turnover. Nevertheless, diminished weight gain was consistent with previous reports [29]; (2) brain histopathological abnormalities were related to Hx and neonatal IH. Persistence of these abnormalities with HiC treatment in IH suggests diminished protection in; (3) oxidative stress in the cerebral cortex was higher with Hx compared to neonatal IH and HiC was more protective; (4) caspase-9, the key initiation step of programmed cell death, was elevated in neonatal IH and caffeine did not prevent it; (5) the known inhibitory effects of caffeine on adenosine receptors appears to be impaired in neonatal IH; (6) when administered in normoxia, caffeine causes hypermyelination, an effect that is abolished in Hx and neonatal IH; and (7) caffeine increases the presence of Golgi bodies in RA, but the effect is diminished in Hx and neonatal IH. Together, these findings imply that caffeine, even at high doses, when administered in normoxia, can confer neuroprotection, evidenced by reductions in oxidative stress, hypermyelination, and increased Golgi bodies. However, varying oxygen environments such as Hx or neonatal IH may alter and modify pharmacodynamic actions of caffeine and may even override the benefits caffeine. On one hand, hyperoxia induction of ROS leading to alterations in redox homeostasis, particularly in immature antioxidant systems, may be responsible for the diminished effects. On the other hand, hypoxia activation of hypoxia-inducible factors (HIFs), and subsequent induction of pathways regulating proliferation, angiogenesis, and invasion, and vessel permeability also contribute to decreases the therapeutic efficacy of caffeine. 

To assess oxidative stress and caspase levels and activity, we biopsied sections from the cerebral cortex, predominantly, layer I. Of the six layers, the most superficial, directly under the pia mater, is the molecular (plexiform) layer, or layer I. It comprises mainly of processes of neurons lying in the deeper layers and their synapses, and only a few horizontal cells. During early postnatal development, the brain undergoes rapid circuit development involving integration of gamma-aminobutyric acid (GABA)ergic neurons into the cerebral cortex [62,63]. IH is a well-established cause of oxidative stress in the brain [64]. In ELGANs with poor nutrition, hypoxia can exacerbate neuronal cell loss and affect GABAergic circuitry and cortical function [65]. It is important to note that samples were analyzed immediately following treatment and Hx or neonatal IH exposure, with no recovery/reoxygenations and thus, we did not assess reperfusion injury. Our findings show that layer I of the cerebral cortex responded to Hx with elevated ROS to a greater extent than neonatal IH and HiC was more effective than LoC for decreasing oxidative stress. This finding is consistent with many reports of the antioxidant capacity of caffeine [56,66,67]. However, this may not be the case for the deeper layers and other areas of the brain. Nevertheless, our results suggest that the immature brain, which is more resistant to hypoxia, is significantly more vulnerable to Hx during the early postnatal period, a time of rapid growth and development. This is also true for IH where there are wide variations from high oxygen to brief low oxygen.

There are three main pathways for oxidative injury, lipid peroxidation of membranes, oxidative modification of proteins, and DNA damage. As a biomarker for oxidative DNA damage, we measured 8-oxo-2′-deoxyguanosine (8-OHdG), an oxidation product generated in DNA through deoxyguanosine oxidation, and is responsible for mutagenesis and carcinogenesis [68]. Caffeine does not appear to have any suppressive effects on 8-OHdG, instead seemed to induce it. This finding supports a previous report of higher urinary 8-OHdG in rat pups fed a coffee diet [69]. Studies on normal quantitative findings of neurons and dendrites in the neonatal rat neocortex are limited. Eayrs and Goodhead [70] reported that by P12, neocortical neurons developed toward inner layer 5b, and the number of dendrites and dendritic elongation reached adult values. Each neuron has an average of 5.4 dendrites, with a maximal length of 18-to-36 μM. In the control hippocampus, neuronal morphologies in all subregions were normal. There was no evidence of neuronal pyknosis, vacuolation, or apoptotic bodies, which are three distinct cellular morphologies of neural death or degeneration. The effects of Hx are devastating to the neocortex and all hippocampal subregions, whereas IH-induced neuronal injuries are limited to the neocortex. Increasing FiO_2_ from 40% to 60% for 12 h and 80% O_2_ for 2 h in immature rat brains significantly increased apoptotic death [71]. Likewise, exposure to 100% O_2_ for 3 h acutely induced neuronal necrosis, possibly through lipid peroxidation [72]. Indeed, Hx is more detrimental to the developing brain than neonatal IH, confirmed in our study.

To further assess cell death, we determined the effects of caffeine and/or Hx or neonatal IH influences caspases. Programmed cell death, apoptosis, is initiated by the formation of an apoptosome in response to several cellular stresses including hypoxia, oxidative stress and DNA damage [73]. Upon triggering of the apoptotic pathways, caspases, a specific family of cysteine proteases, are activated to execute apoptosis [74,75]. Caspase-9 is first initiated and it mobilizes caspase-3 to demolish the cell. Our study showed that IH itself resulted in high levels of caspase-9 in the cerebral cortex homogenates, suggesting activation of the primary step of apoptosis. This finding supports previous data, which demonstrated similar elevations in the hypoxic piglet brain [76]. The caspase-9 to caspase-3 cascade is important and required for apoptosis [77]. However, we did not find subsequent elevations in caspase-3. This may suggest that during the insult, the final stage of apoptosis is not executed. This stage may be activated during recovery/reoxygenation as has been shown to occur during reperfusion injury [78]. Instead, we found lower levels of caspase-3 in all treated groups exposed to Hx and neonatal IH. Caspase-3 has a dual-role during brain development that includes non-apoptosis [79], which promotes dendritic synaptic pruning and wiring, and synaptic strength for transmission or synaptic plasticity [80]. It is also possible that Hx and neonatal IH suppressed apoptotic and non-apoptotic caspase-3 functions. 

Caffeine is a non-selective inhibitor of adenosine receptors, A_1_R and A_2A_R [81] to which its neuroprotective effects are attributed. Adenosine is important for neuroprotection, neuronal and dendritic outgrowth, and controlling synaptic transmission. However, under pathophysiological circumstances, such as hypoxia, extracellular adenosine increases several folds to engage A_1_R and A_2A_R receptors. Activation of A_1_R inhibits excitatory neurotransmitters and synaptic transmission, while A_2A_R activation controls A_1_R overstimulation. Both A_1_R and A_2A_R are highest in the hippocampus and neocortex, but A_2A_R plays the more important roles in the immature brain [82]. It was interesting to note the increased immunoreactivity of A_1_R in response to Hx and neonatal IH, and of A_2A_R in response to neonatal IH suggesting increased excitation in Hx and reductions in neonatal IH. Caffeine seemed to be more selective for inhibiting A_2A_R, than A_1_R and supports previous findings [82], and its inhibitor effect on A_1_R was diminished in neonatal IH. Additionally, of importance, was the area specific induction of A_1_R by HiC in RA which induction in the deeper layers (IV-VI) consisting of stellate, fusiform, and pyramidal cells. Induction of A_2A_R also occurred with HiC treatment in neonatal IH, particularly in the hemorrhagic regions. These findings not only confirm diminished action of caffeine in neonatal IH, but also suggest that chronic use of HiC may have a rebound induction effect resulting in overexpression of A_1_R and A_2A_R, and thus severely interrupting synaptic transmission and neuronal survival. Studies have shown that A_1_R functions as a gatekeeper of neuronal damage, acting as hurdle against noxious insults [83]. Blockade of A_2A_R resulted in robust protection against ischemic brain damage [84]. Therefore, the neuroprotective actions of adenosine may depend on its signaling via A_1_R and bolstering of A_1_R may lead to neuroprotection [83]. Chronic use of caffeine has been shown to induce A_1_R [85], confirming the selectivity of caffeine for A_2A_R inhibition.

The most interesting and important finding of this report is the robust hypermyelination noted with both caffeine doses when administered in RA, suggesting an additional benefit of caffeine, not previously reported. However, previous studies show that caffeine inhibits A_1_R activation to upregulate the expression of myelin-related proteins and promote the morphological differentiation of cultured oligodendrocytes exposed to hypoxia [86]. This same phenomenon may be occurring in our study. In contrast, we found that the pro-myelination effects of caffeine were abolished in Hx and neonatal IH. To our knowledge, this is the first report describing the demyelination effects of Hx and neonatal IH in the developing brain. Golgi stains determine the degree of myelination by evaluating axon impregnation. The Golgi method homogeneously stains neural cell structures, including cell bodies, dendrites, and axons, in the cerebral cortex. This is the also the first report comparing the effects of Hx and neonatal IH, and comparing LoC and HiC on Golgi bodies. In the early postnatal period, when myelination is ongoing, al large portion of the axon is visualized by Golgi stain [87]. Interestingly, we saw reduced staining in the caffeine groups exposed to Hx and a mild increase in neonatal IH. This corresponded with the myelin stains, suggesting that caffeine did not repair Hx- or IH-induced demyelination. This may also indicate that a significant number of neurons may have been severely injured so that the remaining neurons could not launch a compensatory response. 

## 4. Materials and Methods

### 4.1. Experimental Design

All experiments were approved by the State University of New York, Downstate Medical Center Animal Care and Use Committee. Animals were managed according to the guidelines outlined by the United States Department of Agriculture and the Guide for the Care and Use of Laboratory Animals. Certified infection-free pregnant Sprague Dawley rats were purchased from Charles River Laboratories (Wilmington, MA) at 18 days gestation, and remained in a standard environment with food and water provided ad libitum. Sprague Dawley rat pups pooled from 3 litters, were randomly assigned to either: (1) room air (RA); (2) hyperoxia (Hx, 50%O_2_); or (3) neonatal IH (50% O_2_ with brief 1-min episodes of 12% O_2_,) from the first day life (P0) to P14. Within each oxygen environment, pups received either: (1) standard dose caffeine citrate (APP Pharmaceuticals, LLC, Schaumburg, IL USA) administered intraperitoneally (IP) consisting of 20 mg/kg IP on P0 followed by 5 mg/kg/day (LoC) from P1-P14 (*n* = 12; 6 males and 6 females); (2) high dose caffeine (HiC) consisting of 80 mg/kg on P0 and 20 mg/kg/day on P1-P14 (*n* = 12; 6 males and 6 females); or (3) equivalent volume saline (Sal); *n* = 12; 6 males and 6 females (Figure 13, panel A). Gender was determined at birth by the anogenital distance. The pups were weighed on P0, P7, and P14 for caffeine dose adjustment.

### 4.2. Justification for the Doses of Caffeine

Metabolism of caffeine occurs in the liver, and in preterm neonates, ~86% of caffeine citrate is excreted unchanged in the urine [88]. This is due to immaturity of hepatic enzymes that influence half-life of the drug which is longer in neonates than older children and adults. Elimination of caffeine occurs mainly by renal excretion, and this is also slower in premature than older children and adults, because of immaturity of renal function [89]. The use of 100 mg/kg in neonatal rats resulted in increased apoptosis and cell damage in the brain [90]. The doses used in the present study is similar to those administered to preterm infants because of the developmental similarity of the brain between rats at birth and the human preterm infant [56]. Doses of 10 mg/kg in neonatal rats during hyperoxia were shown to be neuroprotective [91].

### 4.3. Intermittent Hypoxia Profile

Pups randomized to Hx or neonatal IH and their mothers were placed into specialized O_2_ chambers (Biospherix Ltd., Lacona, NY, USA) connected to an oxycycler. The O_2_ content inside the chambers was continuously monitored. Changes in O_2_ was adjusted by infusion of nitrogen to decrease O_2_ during IH, and infusion of O_2_ to maintain Hx at 50% O_2_. The IH profile consisted of keeping the rat pups in hyperoxia (50% O_2_) with intermittent bursts of 3 clustered episodes of hypoxia (12% O_2_) each 10 min apart every 2.5 h (Figure 13, panel B). Thus, the neonatal pups were subjected to 8 daily episodes of IH during hyperoxia to simulate a preterm newborn having repeated desaturations during oxygen therapy, as previously reported [49,50,54,55,60,61]. 

### 4.4. Sample Collection

At the end of the experiment, whole brains were removed from the skull and weighed. For ELISA assays, a portion of the cerebral cortex was excised and placed in sterile Lysing Matrix D 2.0 mL tubes containing 1.4 mm ceramic spheres (MP Biomedicals, Santa Ana, CA, USA) and homogenized using a Fast-Prep 24 system (MP Biomedicals, Solon, OH) in ice-cold sterile normal saline (*n* = 6/group). The homogenates were centrifuged at 10,000 rpm for 20 min at 4 °C and filtered prior to assay. A portion of the filtrate was used for total cellular protein assay. For H&E stains, whole brains (*n* = 3/group) were removed, weighed and placed in 10% neutral buffered formalin (NBF). Unstained sections were used immunofluorescence (IF), and myelin staining. For Golgi stains, whole brains were removed and washed in ice-cold double distilled water and placed in containers with Golgi fixative (*n* = 3/group).

### 4.5. 8-isoPGF_2α_, PGF_2α_ and 8-OHdG

8-isoPGF_2α_ is a reliable and proven biomarker for measurement of oxidative stress. It is a byproduct of lipid cell membrane degradation by oxidative stress [92]. Oxidative stress also induces chromatin condensation and lyses DNA, giving rise to high levels of 8-hydroxy-2′-deoxyguanosine (8-OHdG), a specific DNA-fragmented residue. Analysis of 8-isoPGF_2α_, PGF_2α_, and 8-OHdG in the brain homogenates was conducted using commercially available enzyme immunoassays purchased from Enzo Life Sciences (Farmingdale, NY, USA), according to the manufacturer’s protocol. Levels in the homogenates were standardized using total cellular protein levels.

### 4.6. Caspase-3 and -9

Activation of caspase-9 is a key step that initiates execution of programmed cell death in brain following hypoxia. Caspase-3 is the executing molecule involved in apoptotic pathways. Analyses of caspases-3 enzyme activity and caspase-9 levels in the cerebral cortex homogenates were conducted using commercially available enzyme immunoassay kits (MyBiosource, San Diego, CA, USA) according to the manufacturer’s protocol. Levels were standardized using total cellular protein levels.

### 4.7. Total Cellular Protein Levels

On the day of assays an aliquot (10 µL) of the cerebral cortex homogenates, was utilized for total cellular protein levels using the Bradford method (Bio-Rad, Hercules, CA, USA) with bovine serum albumin as a standard.

### 4.8. Histopathology

Whole brains from male and female rats in each group (*n* = 4) were fixed in 10% NBF, cut in 4 mm segments, placed in cassettes, and sent to Histowiz. Inc. (Brooklyn, NY, USA) for processing, sectioning (5 µm), and H&E staining using standard tecniques. Each slide had 3 sections for a total of 12 sections per group. Images were captured at 20× magnification (scale bar = 50 µm) using an Olympus BX53 microscope, DP72 digital camera, and CellSens imaging software (Olympus, Center Valley, PA, USA), attached to a Dell Precision T3500 computer (Dell, Round Rock, TX, USA). Neuronal alignment, cytoarchitecture and morphology of neurons between each group were evaluated by two investigators. Quantitative morphometric analyses were conducted by two investigators using the count and measure tool of the CellSens imaging software. All measurements were conducted in an unblinded manner. 

### 4.9. Myelin Stain

Myelination was determined using the Luxol fast blue stain (VitroVivo Biotech, Rockville, MD, USA). Briefly, unstained cross-sections were de-paraffinized with xylenes and alcohols prior to overnight incubation with Luxol fast blue solution at 56 °C. Slides were washed and differentiated in lithium carbonite solution and 70% ethanol, then counter stained with Cresyl Violet solution, washed and mounted with Permount. Images were captured at 20× magnification (scale bar = 50 µm) using an Olympus BX53 microscope, DP72 digital camera, and CellSens imaging software (Olympus, Center Valley, PA, USA), attached to a Dell Precision T3500 computer (Dell, Round Rock, TX, USA). Stain intensity was determined using the count and measure on region of interest tool of the CellSens imaging software.

### 4.10. Golgi Staining

Golgi stain was used to analyze neuronal morphology, and axonal/dendrite and spines. It is considered to be the most reliable method for analyzing dendritic arborization. After washing in ice-cold double distilled H_2_O to remove the blood elements, the brains were immediately immersed in Golgi impregnation solutions (Hitobiotech, Inc., Wilmington, DE, USA), according to the manufacturer’s recommendations. The stained brain sections were sent to Histowiz, Inc. (Brooklyn, NY, USA) for processing, cutting and mounting on gelatin-coated slides. For staining, the brain sections were placed in a slide warmer overnight, followed by deparaffinization in xylenes and alcohol prior to staining solutions (Hitobiotech, Inc., Wilmington, DE, USA), then dehydrated in alcohol and xylenes prior to mounting. Sections were imaged at 20× magnification (scale bar = 50 µm) using an Olympus BX53 microscope, DP72 digital camera, and CellSens imaging software (Olympus, Center Valley, PA, USA), attached to a Dell Precision T3500 computer (Dell, Round Rock, TX, USA).

### 4.11. Morphometric Analyses

Total brain, layer I (the most superficial layer which lies directly under the pia mater, and comprises few horizontal cells but mainly processes of neurons) and layer II (consists mainly of stellate and small pyramidal cells) widths were determined using the straight-line tool count and measure tool of the CellSens imaging software. For Golgi morphometry, the sections were viewed to examine neuronal distribution under a light microscope at 40× magnification. Neurons and dendrites were more abundant in the neocortex and was our area of interest. Neurons and dendrites in layer III appeared to be the most countable, thus we selected this area to measure: (1) mean survival neurons per section (*n* = 20/group); (2) average length of basal dendrites (measured from the soma to the tip of dendrite, *n* = 25/group); and (3) mean number of dendrites per a neuron (*n* = 25/group) using the count and measure tool of the CellSens imaging software. 

### 4.12. Immunofluorescence (IF) Staining

Unstained brain sections were de-paraffinized and treated with xylenes and ethanol. The slides were placed in 10 mM sodium citrate buffer, pH 6.0 and heated at 95–100 °C for 20 min to unmask the antigens. Following several washes, IF staining was conducted for caspase-3, caspase-9, A_1_R and A_2A_R using primary antibodies purchased from Santa Cruz Biotechnology, Inc. (Dallas, TX, USA) and Alexa Fluor fluorescent secondary antibodies purchased from Life Technologies (Grand Island, NY, USA). All IF protocols were conducted according to the manufacturer’s recommendations. IF sections were imaged at 20× magnification (scale bar = 50 µm) using an Olympus BX53 microscope, DP72 digital camera, and CellSens imaging software (Olympus, Center Valley, PA, USA), attached to a Dell Precision T3500 computer (Dell, Round Rock, TX, USA). Stain intensity was quantified using the count and measure on region of interest tool of the CellSens imaging software. 

### 4.13. Statistical Analyses

Data were analyzed in two ways: (1) comparisons among Sal, LoC, and HiC within each oxygen environment; and (2) comparisons among RA, Hx and neonatal IH between each treatment group. Data for all outcomes are numerica (% change in body weight, brain weight, brain/body weight ratios, 8-isoPGF_2α_, OHdG, caspase-3, caspase-9, PGF_2α_, 8-isoPGF_2α_/PGF_2α_ ratios, and quantitative analyses of caspase-3, caspase-9, A1R, A2AR, and mylenation). A test for normality was first conducted using Bartlett’s test, prior to all statistical analyses. Normally distributed data was analyzed using two-way analysis of variance (ANOVA) with Dunnett’s multiple comparison test. Non-normally distributed data was analyzed using Kruskall Wallis test with Dunn’s multiple comparison test. Data are presented as mean ± SEM and a *p*-value of <0.05 was considered as statistically significant, using SPSS version 16.0 (SPSS Inc., Chicago, IL). Significance differences from Saline RA is presented as * *p* < 0.05 or ** *p* < 0.01; differences from LoC RA is presented as # *p* < 0.05 or ## *p* < 0.01; differences from HiC RA is presented as † *p* < 0.05 or ‡ *p* < 0.05; and differences from placebo saline within each oxygen environment is presented as § *p* < 0.05 or §§ *p* < 0.01.

## 5. Conclusions

Despite these important findings, there are several limitations of this study, which may be the subject of future investigations. First, we did not examine brain responses during recovery/reoxygenation. It is likely that caspase-3, the final step in apoptosis execution may have been elevated as has been shown to occur during reperfusion injury. We completed the experiments at P14 because the human brain growth spurt occurring during the last trimester corresponds to P1-P14 in rats [93]. Second, we did not perform neurological tests to correlate with our findings. Third, we did not utilize the TUNEL staining for detection of apoptosis; and fourth, we did not conduct Sholl analysis on the Golgi images. As stated above, many studies have shown both beneficial and detrimental effects of caffeine on the developing brain. Our findings provide important information that may help resolve these ongoing controversies on standard and high doses of caffeine, as well as provide some guidance regarding the use of HiC and its neurological benefits. Knowledge of the histopathological consequences of chronic exposure to Hx and neonatal IH with HiC treatment in immature neurons is limited. We demonstrated that despite diminished efficacy in Hx and neonatal IH, overall, caffeine treatment resulted in a higher number of neurons and hypermyelination compared to the placebo saline groups. This indicates that caffeine is neuroprotective, promotes neuronal proliferation, and increases dendritic elongation at standard doses. High doses are associated with decreased brain growth and should be avoided unless required clinically for extubation of neonates and weaning from mechanical ventilation. Based on our findings of diminished protection in Hx and neonatal IH, it is recommended to minimize hyperoxic episodes in preterm infants, especially during caffeine therapy. Long-term follow-up studies on histomorphology in animal models together with neurobehavioral evaluations are needed to further establish histological correlates of neurological function and impairment. 

## Figures and Tables

**Figure 1 ijms-22-03473-f001:**
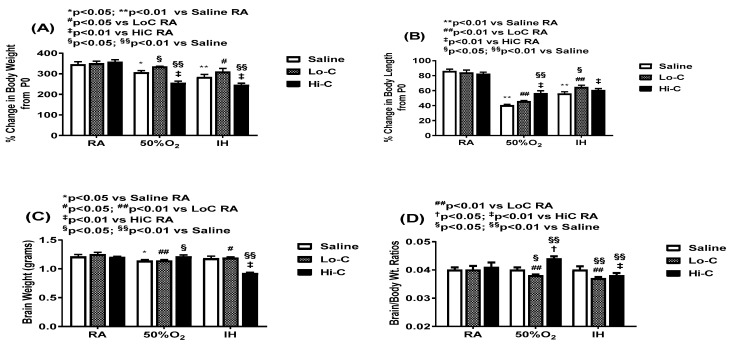
Effect of standard (LoC) and high (HiC) caffeine doses on percentage change in body weight from birth (P0, panel **A**), percentage change in body length from P0 (panel **B**), total brain weight (panel **C**), and brain to body weight ratios (panel **D**). Newborn pups were exposed to: (1) room air (RA); (2) hyperoxia (Hx, 50% O_2_); or (3) neonatal intermittent hypoxia (IH) during which they received IP injections of: (1) LoC (20 mg/kg loading on P0; 5 mg/kg maintenance from P1-P14), grey bars; (2) HiC (80 mg/kg loading; 20 mg/kg maintenance), black bars; or equivalent volume placebo saline from P0 to P14 (white bars). A two-way ANOVA was conducted to examine the effects of oxygen environment (factor 1) and treatment (factor 2) on growth parameters with Dunnett’s 2-sided multiple comparison post hoc test. Saline was used as the control for the treatment groups within each oxygen environment; and room air (RA) was used as the control among the oxygen groups for each treatment. Panel A: F(4,99) = 4.802, *p* < 0.01; Panel B: F(4,99) = 4.437, *p* < 0.01); Panel C: F(4,99) = 14.0, *p* < 0.01; Panel D: F(4,99) = 4.338, *p* < 0.01. * *p* < 0.05, ** *p* < 0.01 vs. Saline RA; # *p* < 0.05, ## *p* < 0.01 vs. LoC RA; † *p* < 0.05, ‡ *p* < 0.01 vs. HiC RA; and § *p* < 0.05, §§ *p* < 0.01 vs. Saline in each oxygen environment. Data are presented as mean ± SEM (*n* = 12 pups/group).

**Figure 2 ijms-22-03473-f002:**
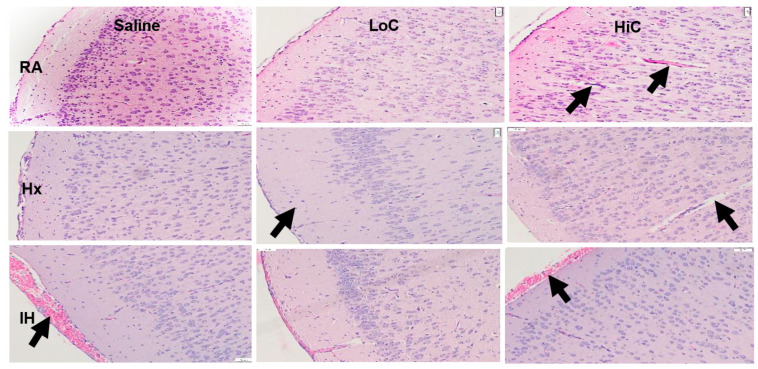
Representative H & E stains of cerebral cortex at P14. Groups are as described in Figure 2. Images are 20× magnification. Scale bar is 50 µM. Groups exposed to placebo saline in RA showed no abnormalities and layers were clearly defined. Hx resulted in loss of cells in the external granular layer (layer II) with reduced distinction of the layers (arrow). Neonatal IH resulted in severe hemorrhage (arrow), but with less reduction in layer II stellate cells. HiC in RA and Hx showed elongated irregular nuclei (arrows). HiC in neonatal IH also showed reduced layering and persistence of hemorrhage (arrow).

**Figure 3 ijms-22-03473-f003:**
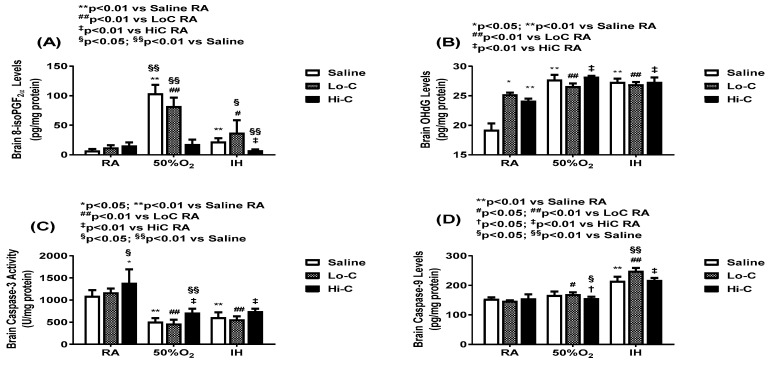
Effect of standard (LoC) and high (HiC) caffeine doses on 8-isoPGF_2α_ (panel **A**), OHdG (panel **B**) and caspase-3 (panel **C**), and caspase-9 (panel **D**) in the cerebral cortex at P14. Groups are as described in Figure 2. A two-way ANOVA was conducted to examine the effects of oxygen environment (factor 1) and treatment (factor 2) with Dunnett’s 2-sided multiple comparison post hoc test. Saline was used as the control for the treatment groups within each oxygen environment; and room air (RA) was used as the control among the oxygen groups for each treatment. Panel A: F(4,45) = 11.251, *p* < 0.01; Panel B: F(4,45) = 17.908, *p* < 0.01; Panel C: F(4,45) = 14.46, *p* < 0.01; Panel D: F(4,45) = 3.135, *p* < 0.01. * *p* < 0.05, ** *p* < 0.01 vs. Saline RA; # *p* < 0.05, ## *p* < 0.01 vs. LoC RA; † *p* < 0.05, ‡ *p* < 0.01 vs. HiC RA; and § *p* < 0.05, §§ *p* < 0.01 vs. Saline in each oxygen environment. Data are presented as mean ± SEM (*n* = 6 samples/group).

**Figure 4 ijms-22-03473-f004:**
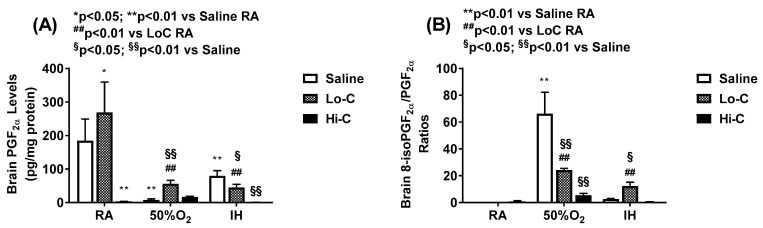
Effect of standard (LoC) and high (HiC) caffeine doses on PGF_2α_ (panel **A**) and 8-isoPGF_2α_/PGF_2α_ ratios (panel **B**) in the cerebral cortex at P14. Groups are as described in Figure 2. A two-way ANOVA was conducted to examine the effects of oxygen environment (factor 1) and treatment (factor 2) with Dunnett’s 2-sided multiple comparison post hoc test. Saline was used as the control for the treatment groups within each oxygen environment; and room air (RA) was used as the control among the oxygen groups for each treatment. Panel A: F(4,45) = 6.4251, *p* < 0.01; Panel B: F(4,45) = 34.985, *p* < 0.01. * *p* < 0.05, ** *p* < 0.01 vs. Saline RA; ## *p* < 0.01 vs. LoC RA; and § *p* < 0.05, §§ *p* < 0.01 vs. Saline in each oxygen environment. Data are presented as mean ± SEM (*n* = 6 samples/group).

**Figure 5 ijms-22-03473-f005:**
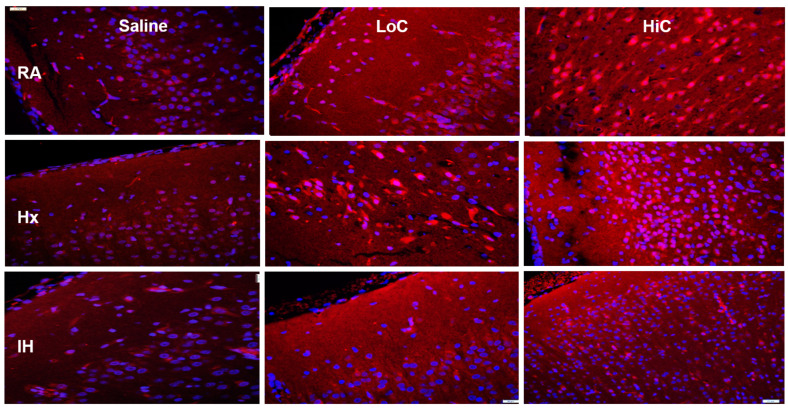
Representative caspase-3 immunofluorescence staining in the cerebral cortex at P14. Groups are as described in Figure 2. Images are 20× magnification, bar scale is 50 µm. Both LoC and HiC resulted in caspase-3 immunoreactivity in all oxygen environments. This may be due to its anti-proliferative effects.

**Figure 6 ijms-22-03473-f006:**
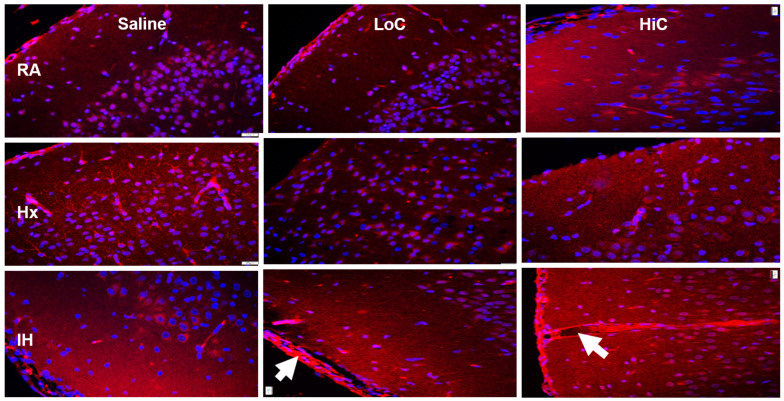
Representative caspase-9 immunofluorescence staining in the cerebral cortex at P14. Groups are as described in Figure 1. Images are 20× magnification, bar scale is 50 µm. LoC in IH showed mild hemorrhage (arrow). HiC induced caspase-9 only in neonatal IH (arrow).

**Figure 7 ijms-22-03473-f007:**
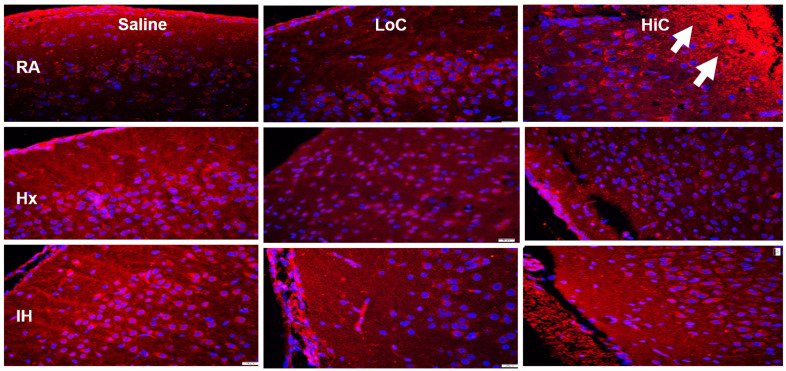
Representative adenosine A1 receptor (A_1_R) immunofluorescence staining in the cerebral cortex at P14. Groups are as described in Figure 2. Images are 20× magnification, bar scale is 50 µm. A_1_R was induced in Hx and neonatal IH in the placebo saline groups, and with HiC in RA (arrow).

**Figure 8 ijms-22-03473-f008:**
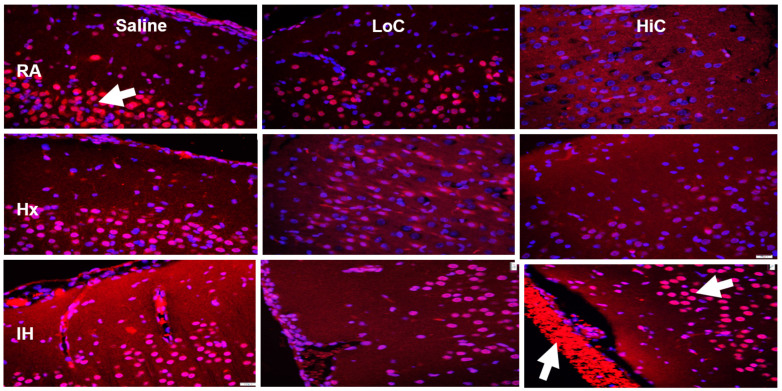
Representative adenosine A2A receptor (A_2A_R) immunofluorescence staining in the cerebral cortex at P14. Groups are as described in Figure 1. Images are 20× magnification, bar scale is 50 µm. A_2A_R was induced in the placebo saline group exposed to neonatal IH, suppressed with LoC in all oxygen environments, and induced with HiC in neonatal IH (arrows). HiC in IH also had severe hemorrhage (arrow).

**Figure 9 ijms-22-03473-f009:**
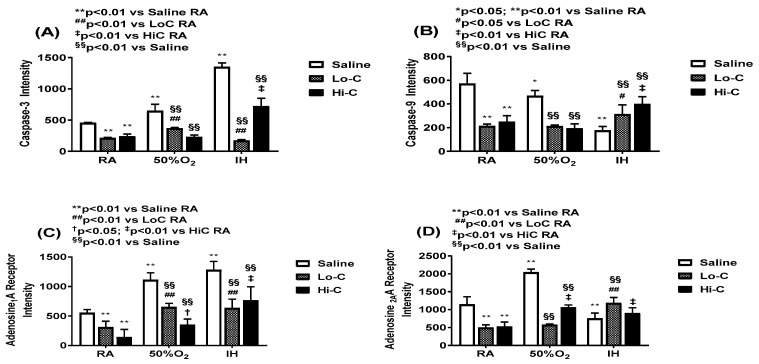
Quantitative analysis of caspase-3 (panel **A**), caspase-9 (panel **B**), A_1_R (panel **C**) and A_2A_R (panel **D**) immunoreactivity presented in Figure 6, Figure 7, Figure 8 and Figure 9. Groups are as described in Figure 1. A two-way ANOVA was conducted to examine the effects of oxygen environment (factor 1) and treatment (factor 2) with Dunnett’s 2-sided multiple comparison post hoc test. Saline was used as the control for the treatment groups within each oxygen environment; and room air (RA) was used as the control among the oxygen groups for each treatment. Panel A: F(4,99) = 7.784, *p* < 0.01; Panel B: F(4,99) = 10.562, *p* < 0.01; Panel C: F(4,99) = 4.029, *p* < 0.01; Panel D: F(4,99) = 7.801, *p* < 0.01. * *p* < 0.05, ** *p* < 0.01 vs. Saline RA; # *p* < 0.05, ## *p* < 0.01 vs. LoC RA; † *p* < 0.05, ‡ *p* < 0.01 vs. HiC RA; and §§ *p* < 0.01 vs. Saline in each oxygen environment. Data are presented as mean ± SEM (*n* = 12 measurements/group).

**Figure 10 ijms-22-03473-f010:**
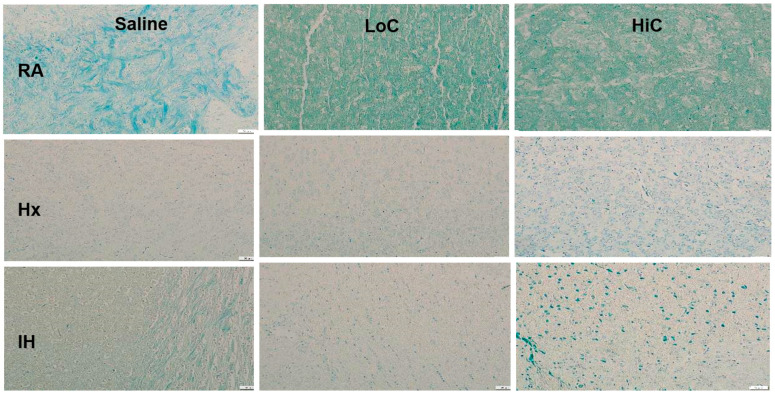
Representative Luxol fast blue staining of myelinated neurons in the cerebral cortex at P14. Groups are as described in Figure 1. Images are 20× magnification, bar scale is 50 µm. Both caffeine doses induced myelination in RA. Hx and neonatal IH suppressed myelination. Treatment with HiC in Hx and neonatal IH mildly increased myelination.

**Figure 11 ijms-22-03473-f011:**
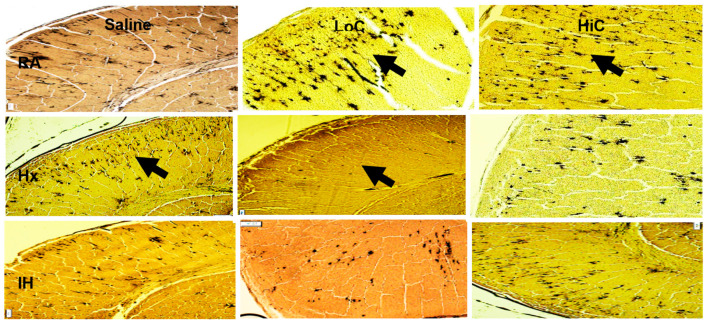
Representative Golgi stain of neurons in the cerebral cortex at P14. Groups are as described in Figure 1. Images are 20× magnification, bar scale is 50 µm. Staining of golgi bodies increased in Hx with placebo saline (arrow), and with LoC and HiC in RA (arrow). However, exposure to caffeine in Hx and neonatal IH decreased golgi staining, consistent with placebo saline.

**Figure 12 ijms-22-03473-f012:**
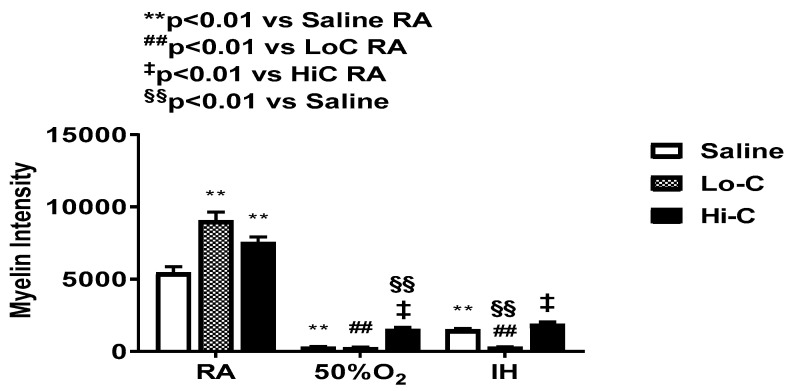
Quantitative analysis of myelin stains presented in Figure 10. Groups are as described in Figure 1. A two-way ANOVA was conducted to examine the effects of oxygen environment (factor 1) and treatment (factor 2) with Dunnett’s 2-sided multiple comparison post hoc test. Saline was used as the control for the treatment groups within each oxygen environment; and room air (RA) was used as the control among the oxygen groups for each treatment. F(4,99) = 20.475, *p* < 0.01. ** *p* < 0.01 vs. Saline RA; ## *p* < 0.01 vs. LoC RA; ‡ *p* < 0.01 vs. HiC RA; and §§ *p* < 0.01 vs. Saline in each oxygen environment. Data are presented as mean ± SEM (*n* = 12 measurements/group).

**Figure 13 ijms-22-03473-f013:**
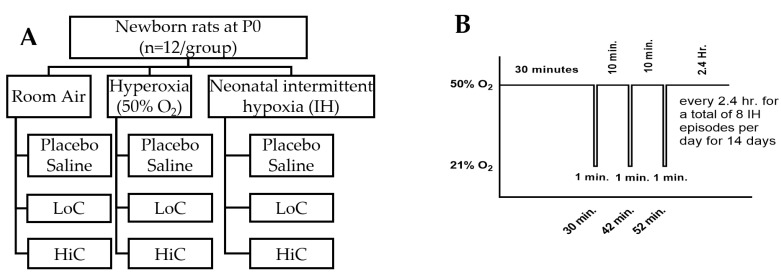
**A**: Experimental Design. LoC (standard caffeine daily IP doses of 20 mg/kg on P0/5 mg/kg on P1-P14). HiC (high caffeine daily IP doses of 80 mg/kg on P0/20 mg/kg on P1-P14). **B**: Neonatal intermittent hypoxia paradigm.

**Table 1 ijms-22-03473-t001:** Blood Gases.

Groups	pH	PCO_2_	PO_2_	SO_2_
**RA:**				
Saline	7.593 ± 0.03	31.1 ± 1.9	83.3 ± 7.9	97.5 ± 0.65
LoC	7.526 ± 0.01	34.0 ± 0.42	69.0 ± 5.6	96.0 ± 0.87
HiC	7.568 ± 0.03	31.7 ± 1.8	122.5 ± 22.5	98.8 ± 0.48
**50% O_2_:**				
Saline	7.584 ± 0.006	37.5 ± 0.49 *	130.5 ± 6.5 **	99.3 ± 0.18 *
LoC	7.58 ± 0.013 #	39.1 ± 1.14 ##	115.5 ± 11.5 ##	98.5 ± 0.61 #
HiC	7.523 ± 0.017 §§	41.8 ± 0.79 ‡§§	88.8 ± 12.7 §	97.0 ± 0.91 §
**Neonatal IH:**				
Saline	7.562 ± 0.037	34.9 ± 2.4	94.0 ± 4.7	98.0 ± 0.41
LoC	7.528 ± 0.02	38.6 ± 1.2 ##	101.5 ± 12.3	98.0 ± 0.71
HiC	7.55 ± 0.026	37.8 ± 1.0 ‡	93.5 ± 20.1	96.8 ± 1.5

RA (room air); LoC (standard dose caffeine—20 mg/kg loading; 5 mg/kg maintenance); HiC (high dose caffeine—80 mg/kg loading; 20 mg/kg maintenance). The p-value following Bartlett’s test for equality of variance was >0.05. A two-way ANOVA was conducted to examine the effects of oxygen environment (factor 1) and treatment (factor 2) on blood gases with Dunnett’s 2-sided multiple comparison post hoc test. Saline was used as the control for the treatment groups within each oxygen environment; and room air (RA) was used as the control among the oxygen groups for each treatment. pH: F(4,99) = 4.598, *p* < 0.01; PCO_2_: F(4,99) = 3.216, *p* < 0.01; PO_2_: F(4,99) = 11.956, *p* < 0.01; SO_2_: F(4,99) = 8.29, *p* < 0.01. * *p* < 0.05, ** *p* < 0.01 vs. Saline RA; # *p* < 0.05, ## *p* < 0.01 vs. LoC RA; ‡ *p* < 0.01 vs. HiC RA; and § *p* < 0.05, §§ *p* < 0.01 vs. Saline in each oxygen environment. Data are mean ± SEM (*n* = 12 measurements/group).

**Table 2 ijms-22-03473-t002:** Brain Morphometric Analyses.

Groups	Total Brain Width (µm)	Layer 1 Width (µm)	Layer 2 Width (µm)
***RA:***			
Saline	6161.5 ± 98.1	129.7 ± 3.6	55.0 ± 2.1
LoC	8534.4 ± 69.5 **	130.7 ± 3.9	57.6 ± 1.6
HiC	8216.7 ± 58.4 **	125.4 ± 7.3	101.0 ± 14.1 **
***50% O_2_:***			
Saline	6864.6 ± 71.9 *	134.9 ± 16.8	104.8 ± 16.1 *
LoC	9166.6 ± 194.1 ##§§	233.1 ± 11.6 ##	119.1 ± 13.6 ##
HiC	7588.9 ± 51.6	114.9 ± 12.8	102.9 ± 11.6
***Neonatal IH:***			
Saline	7240.9 ± 263.9 **	169.8 ± 17.9	120.0 ± 16.3 **
LoC	7905.9 ± 62.5 ##	147.5 ± 17.9	115.7 ± 15.4 ##
HiC	7009.2 ± 212.1 ‡	109.4 ± 14.8 †	105.3 ± 14.2

RA (room air); LoC (standard dose caffeine—20 mg/kg loading; 5 mg/kg maintenance); HiC (high dose caffeine—80 mg/kg loading; 20 mg/kg maintenance). A two-way ANOVA was conducted to examine the effects of oxygen environment (factor 1) and treatment (factor 2) on brain morphometry with Dunnett’s 2-sided multiple comparison post hoc test. Saline was used as the control for the treatment groups within each oxygen environment; and room air (RA) was used as the control among the oxygen groups for each treatment. Total brain width: F(4,99) = 7.72, *p* < 0.01; Layer 1: F(4,99) = 25.47, *p* < 0.01; Layer 2: F(4,99) = 19.109, *p* < 0.01. * *p* < 0.05, ** *p* < 0.01 vs. Saline RA; ## *p* < 0.01 vs. LoC RA; † *p* < 0.05, ‡ *p* < 0.01 vs. HiC RA; and §§ *p* < 0.01 vs. Saline in each oxygen environment. Data are mean ± SEM (*n* = 12 measurements/group).

**Table 3 ijms-22-03473-t003:** Morphologic Parameters of Golgi Bodies.

Groups	Total No. Neurons (*n* = 20 Measurements/Group)	Dendritic LengthµM (*n* = 25 Measurements/Group)	No. Dendrites per Neuron (*n* = 25 Measurements/Group)
***RA:***			
Saline	7.2 ± 0.29	22.3 ± 0.49	2.9 ± 0.2
LoC	22.0 ± 0.43 **	21.9 ± 0.12	4.4 ± 0.02 **
HiC	17.7 ± 0.36 **	12.8 ± 0.08 **	3.8 ± 0.18 **
***50% O_2_:***			
Saline	0.1 ± 0.07 **	0.0 ± 0.0 **	0.0 ± 0.0 **
LoC	2.5 ± 0.3 ##§§	0.04 ± 0.04 ##	0.04 ± 0.04 ##
HiC	8.9 ± 0.26 ‡§§	0.02 ± 0.01 ‡	0.08 ± 0.06 ‡
***Neonatal IH:***			
Saline	4.0 ± 0.25 **	9.9 ± 0.06 **	3.2 ± 0.16
LoC	15.2 ± 0.34 ##§§	14.3 ± 0.12 ##§§	3.5 ± 0.20 ##
HiC	14.4 ± 0.23 ‡§§	11.9 ± 0.06 ‡§§	3.2 ± 0.16 ‡

RA (room air); LoC (standard dose caffeine—20 mg/kg loading; 5 mg/kg maintenance); HiC (high dose caffeine—80 mg/kg loading; 20 mg/kg maintenance). A two-way ANOVA was conducted to examine the effects of oxygen environment (factor 1) and treatment (factor 2) on golgi bodies with Dunnett’s 2-sided multiple comparison post hoc test. Saline was used as the control for the treatment groups within each oxygen environment; and room air (RA) was used as the control among the oxygen groups for each treatment. No. Neurons: F(4171) = 134.59, *p* < 0.01; Dendritic length: F(4216) = 351.269, *p* < 0..01; No. dendrites per neuron: F(4216) = 7.17, *p* < 0.01. ** *p* < 0.01 vs. Saline RA; ## *p* < 0.01 vs. LoC RA; ‡ *p* < 0.01 vs. HiC RA; and §§ *p* < 0.01 vs. Saline in each oxygen environment. Data are mean ± SEM (*n* = 12 measurements/group).

## Data Availability

The data presented in this study are available from the corresponding author upon request.

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
