# Peer review of "Pharmacodynamic Effects of Standard versus High Caffeine Doses in the Developing Brain of Neonatal Rats Exposed to Intermittent Hypoxia"

_ijms, 2021, doi:10.3390/ijms22073473_

Round 1
Reviewer 1 Report
In this study, the authors tested the hypothesis that high-dose caffeine has no adverse effects on the neonatal brain.
I think that the topic may be of interest, and I have some major comments to hopefully improve the text.
- Introduction section it is too short and does not reflect the connection with the announced title. I suggest updating this section.
- for a better understanding of the study methodology, a flow chart would be very good to make the reader better understand the objectives of the study.
- There are many studies on this topic in the literature and as a result, the novelty of the study is missing. I suggest highlighting the novelty of your research: What brings something new to what is already known and proven?
- Because this journal is open access and is addressed to the general public it is necessary to add explanations immediately below figures 2, 5, 6, 7, 8, 10, 12. Not all readers are specialized in histopathology to understand what these images contain.
- Also please list the limitations of this study and the caveats/pitfalls of the study in this cutting-edge topic, as well as the potential aspects limiting the applications of this pharmaceutical compound in “real world” clinical settings.
- While I appreciate that the rate of metabolism is higher in rodents, the calculations commonly used in drug discovery (where the starting values ​​in humans are nano or microMol), the authors must specify using translational medicine the applicability of effective doses from rodents to people, respectively newborns in this case.
Consider revising accordingly!
Author Response
REVIEWER #1:
- We thank the reviewer for his/her comment and we have expanded the Introduction to provide further information and rationale pertaining to the conduct of the study. We have included a number of our own references and specific objectives that relates to the title. Please see pages 1 and 2.
- We have added two figures for further clarity: 1) a graphic abstract that provides a detailed explanation of the mechanisms of caffeine neuroprotection with specific emphasis on the biomarkers used in our study. This is presented as Figure 1 (page 3); and 2) a flow chart of the Study Design and the neonatal intermittent hypoxia (IH) paradigm used in the study are presented as Figure 14, panels A and B, respectively.
- We agree with the reviewer that there are many studies related to the effects of caffeine in the brain and its potential neuroprotection in hypoxia/ischemia. However, there are no studies examining or comparing standard and high doses of caffeine, in the setting of neonatal IH. This is the first study to do this and hence relates to the novelty of the design and findings, pointed out on page 12. Moreover, our model very closely mimics the actual events in newborn babies with neonatal apnea or intermittent hypoxia. This is very important because as we pointed out in the Introduction, there are conflicting reports of high dose caffeine effects in the brain. The main point is the preterm infants, even though accidentally received mega doses, recover with no severe adverse outcomes. However, no studies have examined effects in the brain.
- We have revised the figure legends to add brief explanations regarding the histopathological findings.
- The limitations to the study were stated on page 16 in the “Conclusions” section. However, based on the reviewer’s comments, we have also added the potential limitations of use of caffeine in the clinical setting. Please see page 16, section 5.0.
- We agree with the reviewer that the rate of metabolism is higher in rodents than humans. Metabolism of caffeine occurs in the liver, and in preterm neonates, ∼85% of caffeine is excreted unchanged in the urine [Aldridge A, Aranda JV, Neims AH. Caffeine metabolism in the newborn. Clin Pharmacol Ther. 1979 Apr; 25(4):447-53; Young TE, Mangum B. Neofax: a Manual of Drugs Used in Neonatal Care. Respiratory Drugs. 23rd edn. Montvale, NJ, USA, Thomson Reuters, 2010; pp. 272–273]. This is due to immaturity of hepatic enzymes that influence half-life of the drug which is longer in neonates than older children and adults. Elimination of caffeine occurs mainly by renal excretion, and this is also slower in premature than older children and adults, because of immaturity of renal function [Abdel-Hady H, Nasef N, Shabaan AE, et al. Caffeine therapy in preterm infants. World J Clin Pediatr 2015; 4: 81–93]. Standard dosing regimens result in blood caffeine levels of 6–50 μg/mL in preterm infants. Researchers found that doses of 100 mg/kg achieved similar levels in mice [Kaplan GB, Greenblatt DJ, Kent MA, Cotreau-Bibbo MM. Caffeine treatment and withdrawal in mice: relationships between dosage, concentrations, locomotor activity and a, adenosine receptor binding. J Pharmacol Exp Ther. (1993) 266:1563–73]. Similar high doses were used by Kasala S, et al. (Front Pediatr 2020) in neonatal rats, who found increased apoptosis and cell damage in the brain. We opted to use lower doses that were similar to those administered to preterm infants because a review of animal models of brain injury by Yager and Ashwal (ref. #93) showed that synaptogenesis (cell-cell communication) peaks occurs during the first 3 weeks of life. Since the rat is at the same stage of brain development at 7-10 days as a human at birth, then rat’s brain at birth is developmentally similar to a human preterm infant. Doses of 10 mg/kg in neonatal rats during hyperoxia were shown to be neuroprotective [Endesfelder S, Weichelt U, Strauß E, Schlör A, Sifringer M, Scheuer T, Bührer C, Schmitz T. Neuroprotection by Caffeine in Hyperoxia-Induced Neonatal Brain Injury. Int J Mol Sci. 2017 Jan 18;18(1):187]. We have added these points and references on page 14, section 4.2.
Reviewer 2 Report
To the attention to the authors:
Soontarapornchai and colleagues present an impressive set of experiments where they evaluate the effects of two treatments: one pharmacological (the effects of high and low doses of caffeine) and other physiological (oxygen supply/availability) on the neurodevelopment of the postnatal brain. They establish several levels of analysis, morphological (histopathology of the brain cortex, myelination, morphometry), and physiological (expression of adenosine receptors and levels of blood gases) and toxicological (oxidative stress and apoptosis). The goal was to demonstrate that high doses of caffeine, which is supposed to be neuroprotective against intermittent hypoxia, is no toxic. The work load is impressive because of the richness of approaches and techniques, but my main concern is that the whole history is not well framed. I am going to explain myself:
The tittle states that the goal is to analyze the “pharmacodynamic effects” of high and low doses of caffeine. At the end of the introduction the hypothesis is that high caffeine is not toxic. At the end of the discussion the authors put emphasis in that the most impressive finding is that “the robust hypermyelination noted with both caffeine doses 414 when administered in RA, suggesting” (notice that the sentence is incomplete). And finally the authors say that the effect of caffeine on myelination vanished during intermittent hypoxia. High caffeine has some positive effects but it does not prevent the hemorrhage provoked by the intermittent hypoxia, which is the pathology suffer by neonates. If high caffeine is toxic in animal models and in clinical trials (refs 19 and 20), what is the real goal of the research
A second major concern is the lack of a description of the results of the two-way ANOVAs (by the way, please, indicate what factors are). Neither F values and degrees of freedom of each factor nor the statistical significance of their interactions are provided. The figures use arrows to indicate both hemorrhages and apoptosis. The figure captions do not highlight the main findings and the reader must go to the main manuscript text for details. Only the levels of statistical significance in the figures.
The manuscript requires a thorough substantiation work.
Author Response
REVIEWER #2:
- We thank the reviewer for his/her comments. As stated above, in addition to the overarching objective, we have revised the Introduction to include our specific objectives that are related to the title. Our working hypothesis was that high doses of caffeine confers no adverse effects on the developing brain exposed to neonatal IH. This was based on numerous reports of antioxidant, anti-inflammatory, anti-apoptosis, anti-proliferation, and neuroprotective effects of caffeine.
- We have corrected the incomplete sentence regarding hypermyelination. See page 13.
- We agree with the reviewer that high-dose caffeine in neonatal IH showed evidence of hemorrhage. However, considering that hemorrhage was also shown in the placebo group exposed to neonatal IH, it seems reasonable that the hemorrhage was due to neonatal IH and not the high caffeine dose, suggesting that high doses of caffeine does not prevent IH-induced hemorrhage, but it does not cause it. To further confirm this, we examined many biomarkers of oxidative stress, DNA damage, apoptosis, and neuroinflammation. Similarly, the suppressive effects on myelination was due to neonatal IH and not caffeine. Again, the findings suggest that caffeine does not prevent IH-induced hypomyelination. Taken together, the data suggest that we have proven our hypothesis that high caffeine doses used in this study during neonatal IH, is not neurotoxic. We would like to point out that many of the adverse effects seen in previous studies are likely due to differences in the dose of caffeine used, neurodevelopmental stage at the time of administration, and duration of exposure. Moreover, an overwhelming number of studies have shown the neuroprotective benefits of caffeine in hypoxia/ischemia brain injury.
- The goals of the research are stated in the last paragraph of the Introduction.
- As stated in section 4.13, we analyzed all data by two ways, comparison among the treated groups and comparison among the oxygen groups. All data were numerical (including morphological measurements which were conducted quantitatively) for Figures 3, 5, 6, 11, and 13. No qualitative data are presented. Statistical significance is presented within the figures. However, we have expanded the description of the statistical analyses on page 16, section 4.12.
- We have added explanations of the arrows in the figures and provided a brief summary of the findings in the figure legends.
Reviewer 3 Report
The research paper entitled " Pharmacodynamic Effects of Standard versus High Caffeine Doses in the Developing Brain of Neonatal Rats Exposed to Intermittent Hypoxia" is an interesting piece of work with high scientific value and interest. Authors performed several important qualitative and quantitative parameters to investigate the neuroprotective role of caffeine in developing brain of neonatal exposed to intermittent hypoxia. Working with caffeine is somehow contradictory study as to the best of my knowledge, it shows detrimental effects on cells, however, several interesting data in this paper could be useful to answer questions towards future research possibilities. Although it is an interesting study, there are some limitations in this manuscript that need to be considered before publication. Please find the queries mentioned below.
- Several recent findings are missing in the introduction section, it should be updated.
- Mention the thickness of sections used for qualitative and quantitative study? How did the authors conduct qualitative study? How many sections from each animal were analyzed? What precautionary measures were undertaken to reduce the experimenter bias?
- An inclusion of flow diagram detailing the overall experimental setup could be useful for the readers to understand the study. I will suggest adding a flow diagram in the methods section.
- Authors have mentioned “The pups were weighed on P0, P7, and P14 for caffeine dose adjustment” What was the initial weight of the animals? Without that how can the authors conclude that body weight has reduced in HI/Hx group? The same goes with brain weight.
- In the conclusion section, authors have mentioned several important parameters, which has not been performed in this study as well. Is there any specific reason not to include these parameters in this study? These are important parameters which could be potentially appropriate to signify this study more strongly.
- I will suggest adding a figure defining the mechanisms of the caffeine as a neuroprotectant for neonatal rats by particularly defining the role of parameters performed in this study.
- Indicate all the suggested changes by arrows or an appropriate tool in the figures.
Author Response
REVIEWER #3:
- We thank the reviewer for his/her comment and we have expanded the Introduction to provide more recent findings pertaining to the conduct of the study. Please see pages 1 and 2.
- In the methods section, the thickness of the sections used for the analyses have been added on page 15. Further details regarding the conduct of the analyses, number of sections from each animal, and measures to decrease bias have been included. It should be noted that all data were quantitatively assessed and not qualitatively. Actual measurements were made by two different investigators, who were not blinded to the treatment.
- We have added two figures for further clarity: 1) a graphic abstract that provides a detailed explanation of the mechanisms of caffeine neuroprotection with specific emphasis on the biomarkers used in our study. This is presented as Figure 1; and 2) a flow chart of the Study Design and the neonatal intermittent hypoxia (IH) paradigm used in the study are presented as Figure 14, panels A and B, respectively.
- The initial weights of the pups have been added for each group (page 4). Brain weights were determined only at P14. The effect of caffeine on brain weight and brain to body weight ratios in each group were compared with placebo saline.
- In the conclusions, section 5.0, we stated that that neurological tests, TUNEL staining, and Sholl analysis were not done and were limitations. We agree that these measurements would strengthen our findings. However, neurological examinations in rats require the use of specialized equipment which are not available at our facility. Transport of animals between institutions are prohibited. Sholl analysis is time consuming and requires specific algorithms and technical expertise, that are also not available in our facilities. We opted to use a manual quantitative method to determine dendritic length instead. TUNEL staining is generally used for DNA fragmentation in apoptosis, however, because our model is intermittent hypoxia, we opted to use the OHdG method for oxidative DNA damage, and caspases.
- A figure has been provided to demonstrate the mechanism of caffeine as a neuroprotectant in the neonatal brain, with emphasis on the biomarkers studied. This figure is now a Graphic Abstract – Figure 1.
- We have revised the figures to add arrows to indicate the changes described. We have also added a brief description of the changes in the figure legend.
Round 2
Reviewer 1 Report
No answer given.
Author Response
We thank the reviewer for their review of our revised manuscript.
Reviewer 2 Report
The manuscript has been thoroughly substantiated as required.
However, I still miss the results of the statistic tests (e.g. ANOVAS, ). Where are they? Showing only p-values of the potshoc tests is not enough.
Author Response
We have added the results of the two-way ANOVA in the tables and figure legends.